# Synergistic Effects of Humic Acid, Biochar-Based Microbial Agent, and Vermicompost on the Dry Sowing and Wet Emergence Technology of Cotton in Saline–Alkali Soils, Xinjiang, China

Ge Li [1], Yuyang Shan [1], Yungang Bai [2,*], Weibo Nie [1,*], Qian Wang [1], Jianghui Zhang [2], Hongbo Liu [2], Yu Ding [2], Xiaoyan Wang [2] and Hongqin Lu [2]

1   State Key Laboratory of Eco-Hydraulics in Northwest Arid Region, Xi'an University of Technology, Xi'an 710048, China; lige970901@163.com (G.L.); syy031@126.com (Y.S.); wangqian980223@163.com (Q.W.)
2   Xinjiang Institute of Water Resources and Hydropower Research, Urumqi 830049, China; skyzjh@163.net (J.Z.); lhb090@163.com (H.L.); dy98206@163.com (Y.D.); 17690765020@163.com (X.W.); 18040997591@163.com (H.L.)
*   Correspondence: xjbaiyg@sina.com (Y.B.); nwbo2000@163.com (W.N.)

**Abstract:** Soil amendments such as humic acid (HA), a biochar-based microbial agent (M), and vermicompost (V) can improve soil quality and promote crop growth. However, it remains unclear whether the co-application of the three soil amendments (HMV) has a synergistic effect on alleviating soil quality deterioration obstacles caused by dry sowing and wet emergence technology in Xinjiang cotton fields. A three-year field experiment was conducted in saline–alkali soils using plastic-film-mulched drip irrigation in Xinjiang, China. Through the orthogonal experiment method, the application amounts of HA, M and V were 75 kg ha$^{-1}$, 75 kg ha$^{-1}$ and 225 kg ha$^{-1}$ respectively in 2021. In 2022, three application amount gradients were used for HA, M and V: 60 kg ha$^{-1}$, 90 kg ha$^{-1}$ and 120 kg ha$^{-1}$ respectively. In 2023, the application amounts of HA, M, and V were 60 kg ha$^{-1}$, 120 kg ha$^{-1}$, and 120 kg ha$^{-1}$. It should be pointed out that V contains HA in the range of 20–35%. This study aimed to explore the improvement effect of a single or combined application of HA, M, and V on soil quality and cotton emergence rate using dry sowing and wet emergence technology in Xinjiang cotton fields. The results showed that the single and combined applications of HA, M, and V improved the soil quality and water–heat–salt environment of the cultivated layer. In the combined application, the cotton seedling emergence rate and yield increased by 1.9–22.8% and 7.0–54.1%. Therefore, it is recommended to jointly apply HA, M, and V to promote cotton seedling emergence and increase yield using dry sowing and wet emergence technology in Xinjiang cotton fields.

**Keywords:** soil amendments; dry sowing and wet emergence; soil quality; water–heat–salt environment; yield

## 1. Introduction

As one of the three major cotton-producing areas in China, Xinjiang has made great progress in recent years and has become the main cotton-producing area with the largest output, total output, and planting area in China [1]. As the cotton-planting area in Xinjiang continues to increase, agricultural water accounts for more than 97%, and irrigation water is increasingly scarce in the winter and spring [2]. Poor thermal conditions and insufficient heat accumulation in Xinjiang are the main factors limiting cotton production. There is a shortage of thermal resources during the cotton seedling stage [3]. Cotton is susceptible to cold damage during the seedling stage as well as rapid cooling later in life, which makes it vulnerable to frost in the fall [4]. In view of the climate characteristics of insufficient thermal resources in this area, a series of early stimulus measures were taken to accelerate

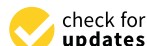



the growth and development process of cotton so that most cotton fields could meet the growth needs. Therefore, the dry sowing and wet emergence technology is widely implemented in this area, that is, neither winter nor spring irrigation is required before cotton sowing, and mulching film and drip irrigation tape are directly laid for sowing after soil preparation. Once the optimal emergence temperature is reached, drip irrigation with small amounts of water allows for the seeds to emerge [5,6].

Compared with conventional winter and spring irrigation sowing technology, dry sowing and wet seeding technology has a higher seedling emergence rate and uses only about 20% of the water, which greatly alleviates the pressure caused by water shortage in the region [3]. In terms of soil temperature, dry sowing and wet seeding can effectively increase and stabilize the surface soil temperature, causing the peak temperature of the bottom soil to lag behind the surface soil temperature. The cooling rate of the surface soil is higher than that of the bottom soil, promoting the rapid germination of cotton [6]. Combined with the mature cotton under-film drip irrigation technology commonly used in Xinjiang [7], cotton dry sowing and wet emergence technology has high promotion value. It is worth noting that although the dry sowing and wet emergence technology saves irrigation water, it also causes salt accumulation in the soil in the cultivated layer due to the small amount of irrigation water [8]. Therefore, improving the continued deterioration of soil conditions in Xinjiang cotton fields is crucial to promoting the sustainable development of the agricultural production system.

Humic acid (HA) is a supramolecular structure. It can be extracted from the used materials such as vermicompost (V), can be used as a soil amendment, and has a better promotion effect on the growth of crop roots [9–11]. Zhang et al. [12] found that HA itself can be used as an organic fertilizer to improve soil fertility. Li et al. [13] found that HA increased soil nutrient content, including total nitrogen, total phosphorus, total potassium, available nitrogen, available phosphorus, available potassium, and organic matter. As a soil conditioner, the acidic functional groups in the molecular structure of HA can dissociate hydrogen ions and react with polyvalent cations (such as $Ca^{2+}$, $Mg^{2+}$, $Al^{3+}$, etc.) on the soil surface to form a complex. This complexation can change the activity and solubility of metal ions in soil and reduce the toxicity of metals. At the same time, the complexation of HA can also promote the release and supply of nutrients, improve soil fertility, and promote the growth and development of plants [14]. Secondly, HA has a large number of hydroxyl, carboxyl, and other functional groups in HA, so it can hydrogen bond with water molecules to form a solution. This hydrophilicity allows HA to promote the formation of soil aggregates [15], inhibit soil evaporation [16], and accelerate the leaching and irrigation of salts and alkali on the soil surface [17]. Hu et al. [18] studied that adding HA to soil can reduce soil conductivity, water-soluble $Na^+$ and $K^+$ content, and sodium adsorption rate, and significantly increase cotton yield.

The biochar-based microbial agent (M) is prepared by immobilizing microbial inoculants on biochar [19]. Currently, it is rarely used in saline soil remediation. Nguyen et al. [20] showed that biochar and microbial interactions have the potential to increase soil nutrient content, water retention, hydrocarbon biodegradation, soil fertility, and inertia-induced plant disease resistance. Qi et al. [19] prepared M by immobilizing microbial agents containing Bacillus subtilis, Bacillus cereus, and Citrobacter on biochar and adding them in the soil. It was found that the soil organic matter increased by 58.7%, which greatly promoted the growth of vegetables. Pang et al. [21] found that adding M to the soil can reduce the total salinity and pH value of the topsoil layer, which is beneficial to the accumulation of dry matter in the upper ground of corn. Wu et al. [22] found that adding M to the soil can increase the total organic carbon content of the soil and promote the accumulation of dry matter in the roots, stems, leaves, and grains of rice during the maturity stage. From the perspective of improving soil quality, M can be used as an ideal soil repair amendment. However, compared with the effects of M application on the toxic heavy metals uranium and cadmium, the effects of M application on soil physical and hydrothermal properties have received less attention.

Applying V to soil can improve soil physical, chemical, and biological properties, effectively reduce soil continuous cropping obstacles in the field, and improve crop growth, productivity, and quality [15,23]. The humus in V droppings and calcium-ion-cemented soil particles are condensed to form a granular structure, which has strong water retention performance [24]. The improvement of soil chemical properties of V is mainly reflected in increasing nitrogen, phosphorus, potassium, and organic matter content, improving soil nutrient state, promoting crop nutrient absorption, and affecting soil fertility and crop growth [25]. Wang et al. [26] found that the application of V significantly reduced soil electrical conductivity and improved cucumber yield and quality. The combined effect of V and biochar performs better in reducing electrical conductivity and increasing dissolved organic carbon.

Recently, there has been increasing interest in the synergistic effects of soil amendments, and Doan et al. [27,28] found that the combined effect of V and biochar can increase soil stability and reduce soil organic carbon content. Alvarez et al. [29] found that the combined effect of V and biochar benefits crop productivity while reducing the negative impact of agriculture on ammonia and nitrous oxide emissions. Han et al. [30] found that the combined effect of biochar and HA was more obvious in reducing soil pH, conductivity, and alkalization, and increasing soil nutrients. The nutrient content of HA is relatively high, and it can promote the activity of soil microorganisms and increase the number of beneficial microorganisms in soil, but its microbial species and quantity are small. Secondly, the nutrient content of V is low, but its good physical and biological characteristics can promote the formation of soil aggregate structures, reduce the leaching of nutrients, accelerate the decomposition and transformation of soil organic matter, and thus increase the content of soil available nutrients [26]. Finally, the M contains a large number of beneficial microorganisms, which can inhibit pathogens in the soil and reduce the occurrence of soil-borne diseases [22]. Considering that HA, M, and V can all have the advantage of improving degraded soil, their synergistic effects can provide practical ways to improve soil quality and increase sustainable cotton production.

Therefore, this study takes cotton in Xinjiang as the research object and aims to determine whether the coordinated application of HA, M, and V (HMV) has a synergistic beneficial effect on cotton seedling emergence using dry sowing and wet emergence technology. Considering the research gaps, this study hypothesizes that (1) HMV improves soil physical and chemical properties by increasing soil porosity, soil water content, and soil temperature, and reducing soil bulk density, soil salt content, and soil compaction degree, and that (2) improving soil properties will directly and indirectly promote cotton emergence rates.

## 2. Materials and Methods

### 2.1. Study Site and Initial Soil Properties

This study conducted a three-year field experiment in the cotton fields of Hailou Town, Shaya County, Aksu City, Xinjiang Uygur Autonomous Region (82°71′ E, 41°28′ N, altitude 985 m) from 2021 to 2023 (Figure 1). The study site has a typical continental warm temperate arid climate, with a maximum potential evaporation of 2072.6 mm and an average annual rainfall of 64.9 mm. Meteorological data such as wind speed, air temperature, precipitation, and relative humidity were measured at the test site using a portable, small automatic weather station (HOBO U30, MA, USA). In the cotton-growing seasons of 2021, 2022, and 2023, the average temperatures were 21.8 °C, 22.0 °C, and 20.8 °C, respectively; the precipitations were 27.6 mm, 38.2 mm, and 55.4 mm, respectively; and the average wind speeds were 2.9 m s$^{-1}$, 2.7 m s$^{-1}$ and 3.4 m s$^{-1}$ respectively (Figure 2).

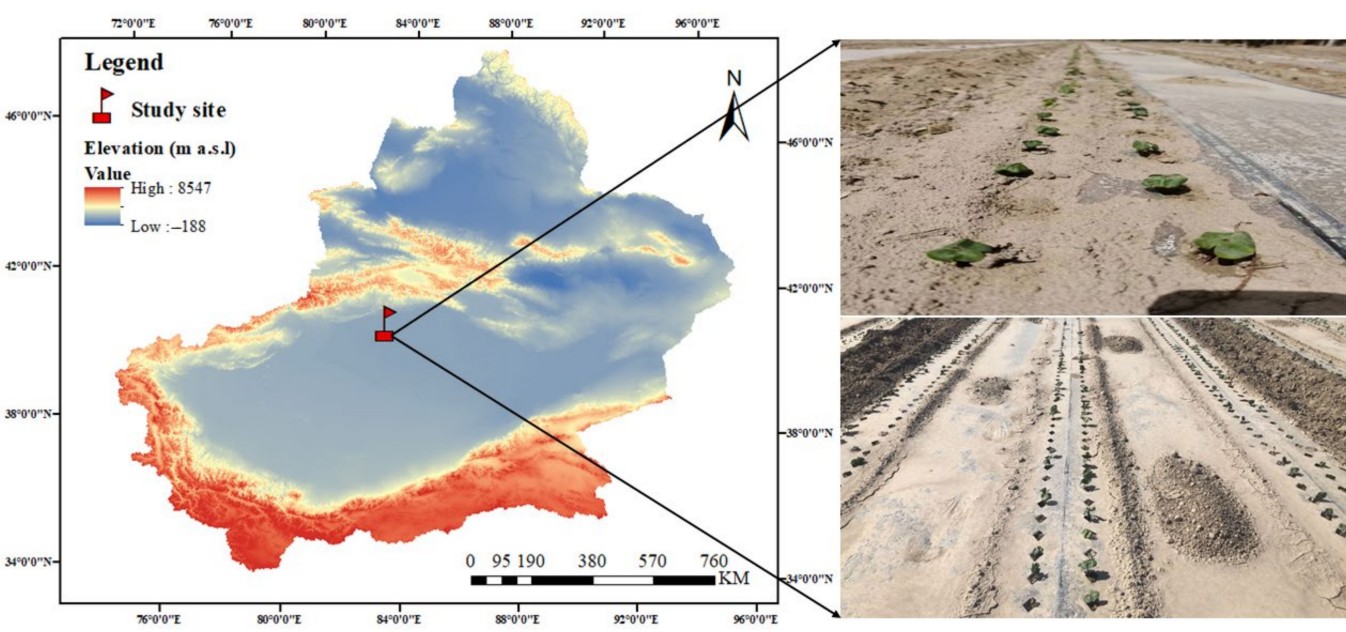

**Figure 1.** Experimental area and location of sampling points.

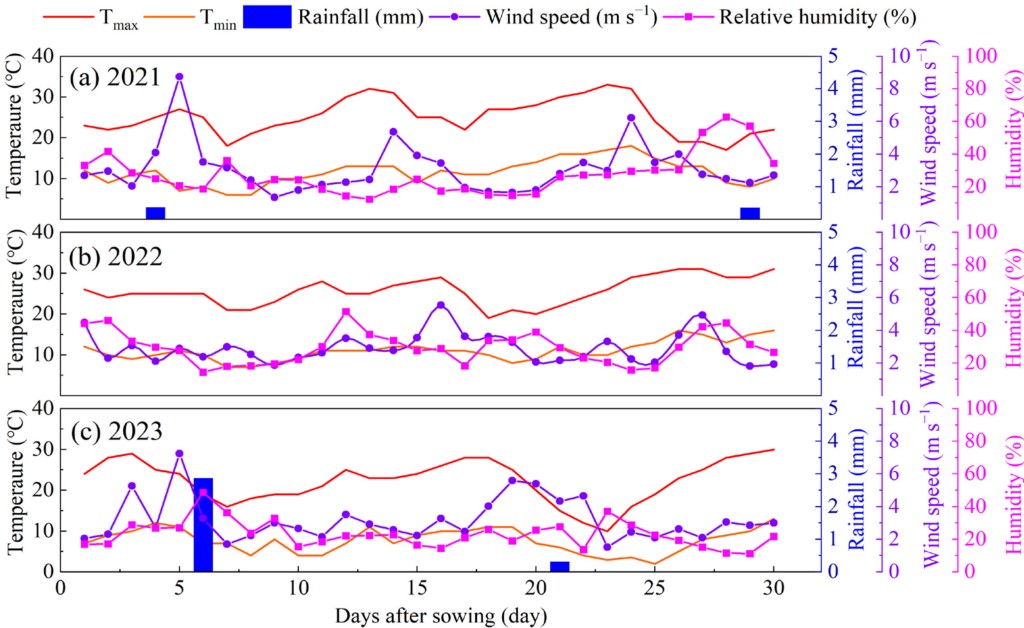

**Figure 2.** Meteorological data during the period of dry sowing and wet emergence of cotton from 2021 to 2023. $T_{max}$ and $T_{min}$ are the maximum and minimum air temperature, respectively.

In 2021, before the start of the trial, pre-seeding soil samples were collected at a depth of 0 to 40 cm. Malvern laser particle size analyzer (Mastersizer 2000) was used to measure the content of clay, silt, and sand at different soil depths (0–10 cm, 10–20 cm, 20–30 cm, and 30–40 cm). According to the soil texture classification standards of the United States Department of Agriculture, soils with a depth of 0–40 cm are classified as silt loam (Table 1).

**Table 1.** Physical and chemical properties of the experimental soils in 0–40 cm soil layer.

| Soil Layer (cm) | SBD (g cm$^{-3}$) | $\theta_s$ (cm$^3$ cm$^{-3}$) | FC (cm$^3$ cm$^{-3}$) | Sand (%) | Silt (%) | Clay (%) | EC$_{1:5}$ (mS cm$^{-1}$) | PH |
|---|---|---|---|---|---|---|---|---|
| 0–10 | 1.48 | 0.352 | 0.295 | 35.7 | 55.2 | 9.1 | 0.36 | 8.7 |
| 10–20 | 1.46 | 0.346 | 0.284 | 41.2 | 53.0 | 5.8 | 0.26 | 8.7 |
| 20–30 | 1.45 | 0.356 | 0.294 | 39.9 | 51.3 | 8.8 | 0.23 | 8.7 |
| 30–40 | 1.50 | 0.345 | 0.282 | 36.9 | 55.0 | 8.1 | 0.27 | 8.7 |

SCL represents silt clay loam; SBD represents bulk density; $\theta_s$ represents saturated soil water content; FC represents field capacity.

## 2.2. Field Experimental Design

### 2.2.1. Sources of HA, M, and V

HA: produced by Xingnong Pharmaceutical (China) Co., Ltd., Shanghai, China; dark brown liquid containing 10% potassium humate; organic matter $\geq$ 40.0%; humic acid $\geq$ 60 g L$^{-1}$; total nutrients (N + P$_2$O$_5$ + K$_2$O) $\geq$ 200 g L$^{-1}$. M: produced by Yunnan Tianjuli Fertilizer Co., Ltd., Yunnan, China; number of effective viable bacteria $\geq$ 5 billion g$^{-1}$; organic matter $\geq$ 50.0%; organic carbon base $\geq$ 25%; cellulase $\geq$ 3000 mg kg$^{-1}$; V: produced by Xinjiang Zhongnong Hongyuan Agricultural Technology Co., Ltd., Xinjiang, China; organic matter $\geq$ 24.5%; humic acid 20–35%; crude protein 5.0–9.0%; total nutrients (N + P$_2$O$_5$ + K$_2$O) $\geq$ 25%; pH around 5.3; beneficial viable bacteria count 2.2 billion/g fresh weight.

### 2.2.2. Treatment of HA, M, and V

The 2021 trial adopted a three-factor, single-level design and a control group with a total of 8 treatments. The application rates of HA, M, and V were 75 kg ha$^{-1}$, 75 kg ha$^{-1}$, and 225 kg ha$^{-1}$, marked as CK, HA, M, V, HM, HV, MV, and HMV, respectively. Each treatment was repeated three times. Each plot area was 42 m$^2$. A completely randomized block design was adopted. A 1.5 m-wide channel was set up between adjacent plots to avoid cross-contamination caused by irrigation and fertilization. HA, M, and V were fully mixed with water and applied to the soil with water by drip irrigation. The irrigation occurred once and the irrigation quota was 900 m$^3$ ha$^{-1}$.

In 2022, a total of 10 treatments were used, including a three-factor, three-level orthogonal test and a control group. Each treatment was repeated three times. The three factors were HA, M, and V. The three application amounts were 60 kg ha$^{-1}$, 90 kg ha$^{-1}$, and 120 kg ha$^{-1}$, marked as CK, HMV1, HMV2, HMV3, HMV4, HMV5, HMV6, HMV7, HMV8, and HMV9.

In 2023, the application amounts of HA, M, and V were 60 kg ha$^{-1}$, 120 kg ha$^{-1}$, and 120 kg ha$^{-1}$, respectively, marked HMV10 (Table 2). In 2021, 2022, and 2023, the base fertilizer applied before ploughing was urea (N $\geq$ 46%) 450 kg ha$^{-1}$, diammonium phosphate (N $\geq$ 46% and P$_2$O$_5$ $\geq$ 46%) 300 kg ha$^{-1}$, and potassium sulfate (K$_2$O $\geq$ 52%) 150 kg ha$^{-1}$.

**Table 2.** Treatments of experiment.

| Year | Treatment | Application Amount (kg ha$^{-1}$) | | |
|---|---|---|---|---|
| | | Humic Acid | Biochar-Based Microbial Agent | Vermicompost |
| 2021 | CK | 0 | 0 | 0 |
| | HA | 75 | 0 | 0 |
| | M | 0 | 75 | 0 |
| | V | 0 | 0 | 225 |
| | HM | 75 | 75 | 0 |
| | HV | 75 | 0 | 225 |
| | MV | 0 | 75 | 225 |
| | HMV | 75 | 75 | 225 |

**Table 2.** *Cont.*

| Year | Treatment | Application Amount (kg ha$^{-1}$) | | |
|---|---|---|---|---|
| | | Humic Acid | Biochar-Based Microbial Agent | Vermicompost |
| 2022 | CK | 0 | 0 | 0 |
| | HMV1 | 60 | 60 | 60 |
| | HMV2 | 90 | 90 | 60 |
| | HMV3 | 120 | 120 | 60 |
| | HMV4 | 90 | 60 | 90 |
| | HMV5 | 120 | 90 | 90 |
| | HMV6 | 60 | 120 | 90 |
| | HMV7 | 120 | 60 | 120 |
| | HMV8 | 90 | 90 | 120 |
| | HMV9 | 60 | 120 | 120 |
| 2023 | CK | 0 | 0 | 0 |
| | HMV10 | 60 | 120 | 120 |

### 2.2.3. Cotton Cultivation and Planting Pattern

The cotton variety Yuanmian No. 11 was used as the material. The sowing times were 18 April 2021, 10 April 2022, and 14 April 2023, and the sowing density was $2.6 \times 10^5$ plants ha$^{-1}$. The planting method and dropper layout of the cotton fields were based on the local model of "one plastic film, three drip lines, and six rows of cotton plants" (Figure 3). Three drip lines were located beneath one plastic mulch, and six rows of cotton plants were covered by a plastic mulch. Cotton plants were spaced 10 cm in each row, and each dropper consisted of drippers spaced 30 cm apart. The discharge rate of each emission tube was 2.0 L h$^{-1}$. The wide and narrow row areas were determined by the position of the cotton plant. The wide and narrow row widths were 66 cm and 10 cm, respectively.

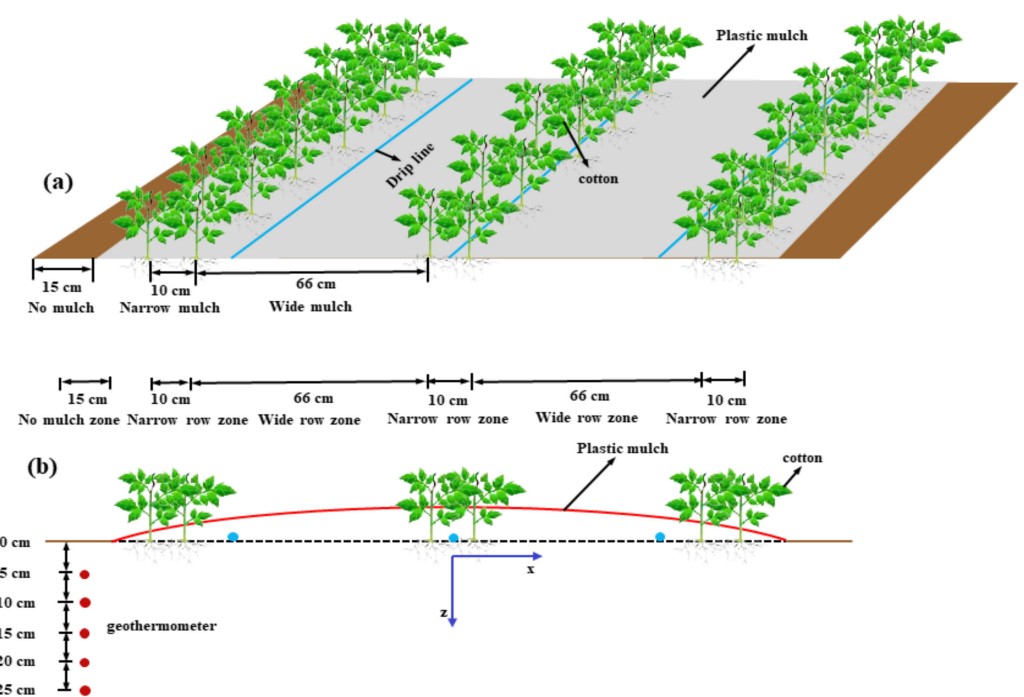

**Figure 3.** Schematic diagram of cotton-planting pattern, drip line arrangements, one mulch, three drip lines, and six rows of cotton plants. (**a**) represent diagram of test layout; (**b**) represent schematic diagram of vertical soil section.

*2.3. Measurement Indicators and Methods*

2.3.1. Soil Bulk Density and Porosity

Soil bulk density (SBD) was measured by the cutting ring method. Before sowing in 2021, 2022, and 2023, soil profiles 20 cm-deep were excavated in the test plot, and a ring knife with a volume of 100 cm$^3$ was inserted into the 0–20 cm soil layer of each test plot profile to measure SBD. The soil porosity (SP) calculation formula is as follows [31]:

$$SP = 1 - \frac{SBD_m}{SBD_p} \tag{1}$$

where $SBD_m$ is measured soil bulk density (g cm$^{-3}$) and $SBD_p$ is soil particle density (=2.65 g cm$^{-3}$).

2.3.2. Soil Water Content

Soil water content was determined by the drying method. In 2021, soil samples were collected at 10, 20, 30, and 40 cm soil layers using an auger (diameter 5 cm, length 15 cm) in uncovered, narrow row, and wide row areas. Soil samples were collected in 2022 and 2023 at 10, 20, 30, and 40 cm soil layers in narrow and wide row areas. A portion of the collected soil samples were dried at 105 °C until they reached a constant weight to determine the gravimetric soil water content. Soil volumetric water content (SWC) is obtained by multiplying gravimetric water content and bulk density.

2.3.3. Soil Salt Content

Soil salt content was determined by the conductivity method. First, 0–10, 10–20, 20–30, and 30–40 cm soil samples were collected 15 days after cotton sowing, air-dried, ground into fine pieces, and then 20 g was weighed and placed in a triangular bottle with a volume of 250 mL through a 1 mm screen. The leaching solution with a soil–water ratio of 1:5 was prepared, and the leaching solution was oscillated on an oscillator for 15 min, allowed to stand still for 30 min, and then filtered. The conductivity value of the leaching solution was measured ($EC_{1:5}$) using a DDS-307 conductivity meter (DDS-307, INESA Scientific Instrument Co., Ltd., Shanghai, China). Then, the soil salt content (SSC) was obtained through the relationship between electrical conductivity and SSC, as shown in Figure S1.

2.3.4. Soil Temperature

Easylog-USB automatic monitoring ground thermometer (Lascar, Inc., Whiteparish, UK) was used to monitor the soil temperature (ST). The ground thermometer was buried between the two rows of crops under the film and directly under the drip irrigation belt at soil depths of 5 cm, 10 cm, 15 cm, 20 cm, and 25 cm. The instrument was started in mid-April. Data monitoring, continuous monitoring throughout the emergence period, and soil temperature monitoring every 30 min were carried out.

2.3.5. Soil Compaction Degree

After cotton emerged in 2021, 2022, and 2023, 10 groups of cotton cavities were randomly selected from each treatment point, and the PC40 II soil compaction meter was used to measure the soil compaction degree (SCD) in the 0–20 cm soil layer.

2.3.6. Cotton Seedling Emergence Rate

Fifteen days after sowing, the cotton seedling emergence rate (the number of seedlings emerging as a percentage of the total number of seeds sown) at the sample points (three films in the middle) of each treatment was counted. In total, 100 holes of cotton seedlings were measured at each sample point to compare the emergence rates (SERs) between treatments.

### 2.3.7. Cotton Yield

A 6.67 m$^2$ production measurement area was defined in advance in each plot of the cotton field. During the cotton harvest period, seed cotton was collected manually and air-dried in the laboratory to measure the seed cotton yield (CY).

### 2.4. Statistical Analyses

One-way analysis of variance (ANOVA) was performed using SPSS 17.0 software. The least significant difference (LSD) at the $p < 0.05$ level was used to determine the significant difference between treatments. One-factor analysis of variance was used to test the effects of HA, M, and V on SBD, SP, SWC, ST, SCD, SSC, SER, and CY. Origin 2018, Microsoft Office Excel 2019, and Microsoft Office PowerPoint 2019 were used for drawing and data analysis.

In order to achieve the research objectives, the research framework followed by this study is shown in Figure 4.

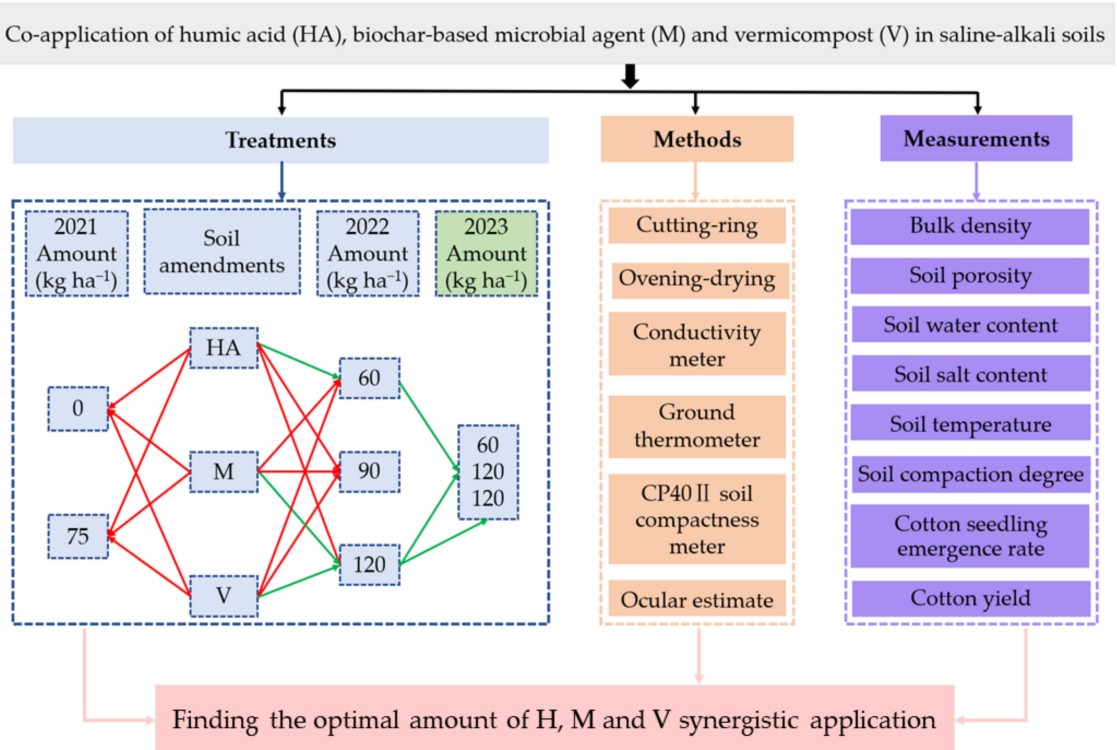

**Figure 4.** The research framework in this study.

## 3. Results

### 3.1. Effects of HMV Application on Soil Bulk Density and Porosity

In 2021, 2022, and 2023, compared with the control, SBD decreased by 2.2–20.8% in the HA, M, and V application, and SP increased by 4.4–87.1% (Figure 5). The 2021 test results showed that the combined application of HA, M, and V had a better effect on SBD and porosity than the single applications of HA, M, and V. In the combined application of HA, M, and V in 2022, the application amounts of HA, M, and V had a great impact on SBD and SP. The application amounts of HMV9 had the best improvement effect on SBD and SP, followed by HMV8. We found that SBD and SP had the best improvement effects when the application amounts of HA, M, and V were 60 kg ha$^{-1}$, 120 kg ha$^{-1}$, and 120 kg ha$^{-1}$, respectively. The field practice in 2023 found that the optimal application rate of the combined application of HA, M, and V in 2022 still had a great improvement effect on SBD and SP. Over three years of field experiments, it was found that the three amendments improved soil physics, and their combined application effect was better.

The optimal combined application amounts of HA, M, and V were 60 kg ha$^{-1}$, 120 kg ha$^{-1}$, and 120 kg ha$^{-1}$.

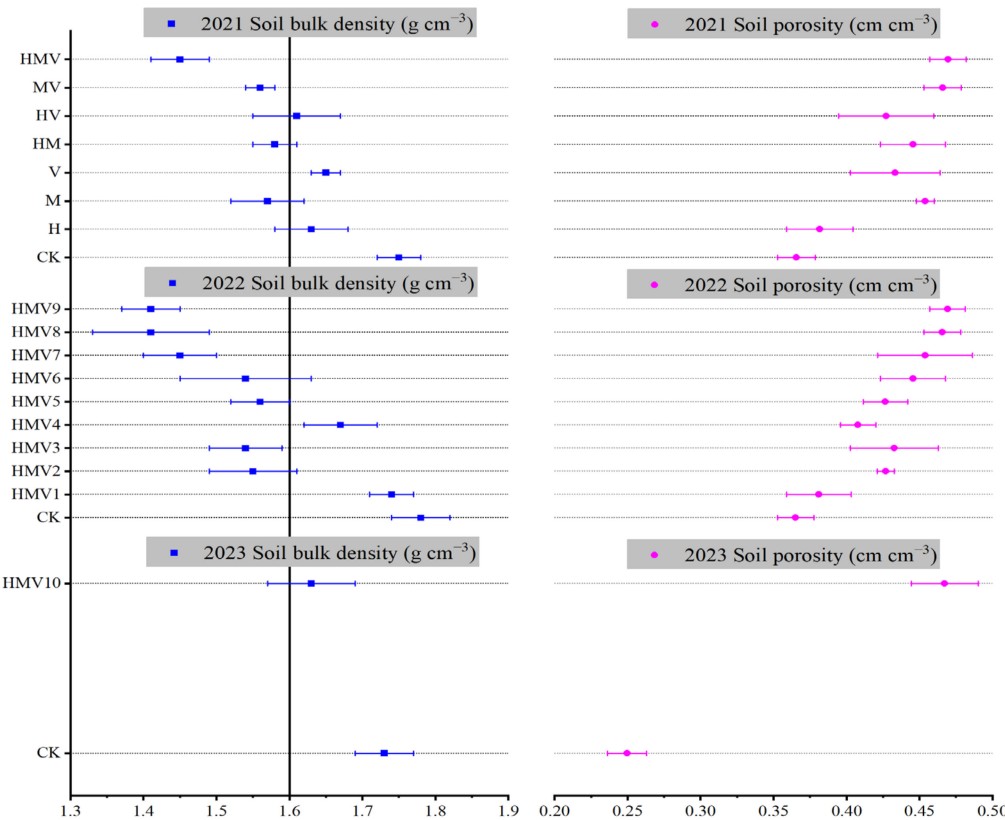

**Figure 5.** The effects of different HMV treatments on 0–20 cm bulk density and porosity of 0–20 cm cotton dry sowing and wet emergence stages in 2021, 2022, and 2023. Error bar represents the standard error.

### 3.2. Effects of HMV Application on Soil Water Content, Soil Temperature, and Soil Salt Content

3.2.1. Soil Water Content

Under film-mulched drip irrigation conditions in 2021, 2022, and 2023, the application of HA, M, and V had an impact on the distribution of SWC in the 0–40 cm soil layer at the dry sowing and wet emergence stages of cotton (Figures 6–8). To simplify the analysis, the changes in SWC with soil depth in the HA, M, and V treatments in 2021, 2022, and 2023 were compared (Figures S2–S4). The results showed that compared with the control group, the average SWC of the 0–40 cm soil layer in the HA, M, and V treatments in 2021 increased by 11.1–31.9%. Under the synergistic effect of HA, M, and V, the average SWC of the 0–40 cm soil layer increased by 2.0–45.9% in 2022 and 3.3% in 2023. It showed that the application of HA, M, and V improved the water retention capacity of cotton shallow tillage soil and ensured the soil water demand for cotton seedlings.

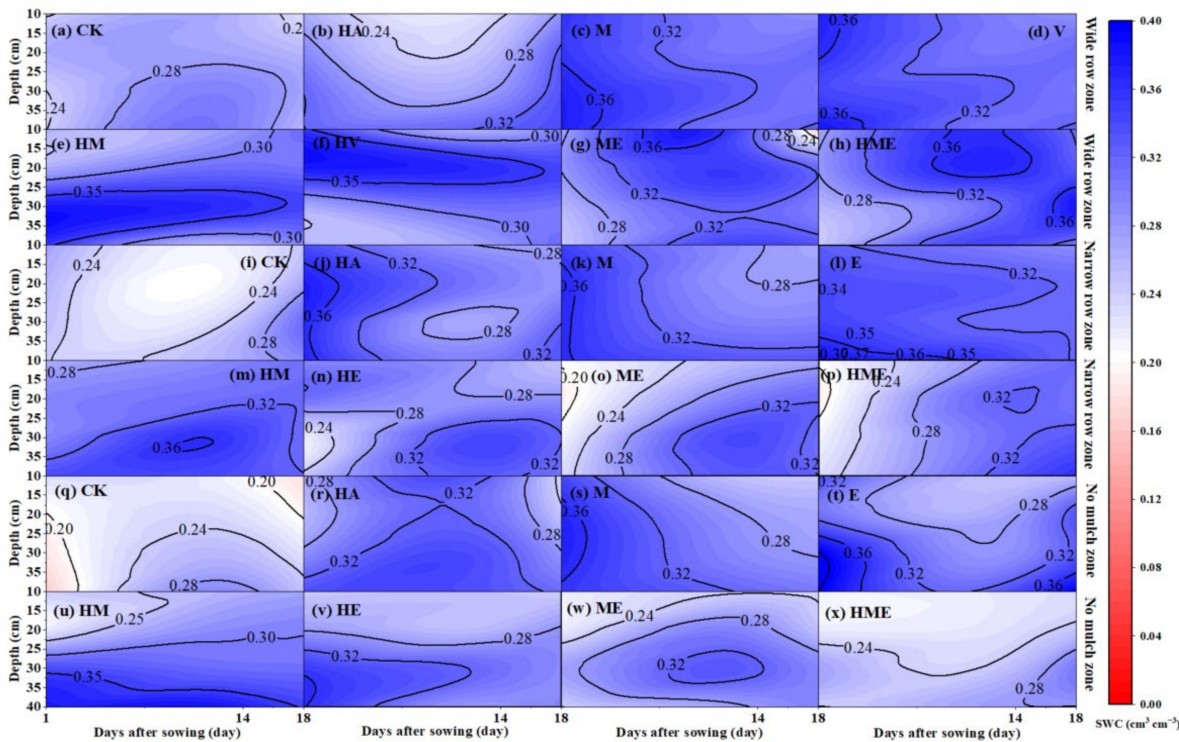

**Figure 6.** Distribution of soil water content in soil using film-mulched drip irrigation at dry sowing and wet emergence stages for the humic acid (HA), biochar-based microbial agent (M), and vermicompost (V) treatments in 2021.

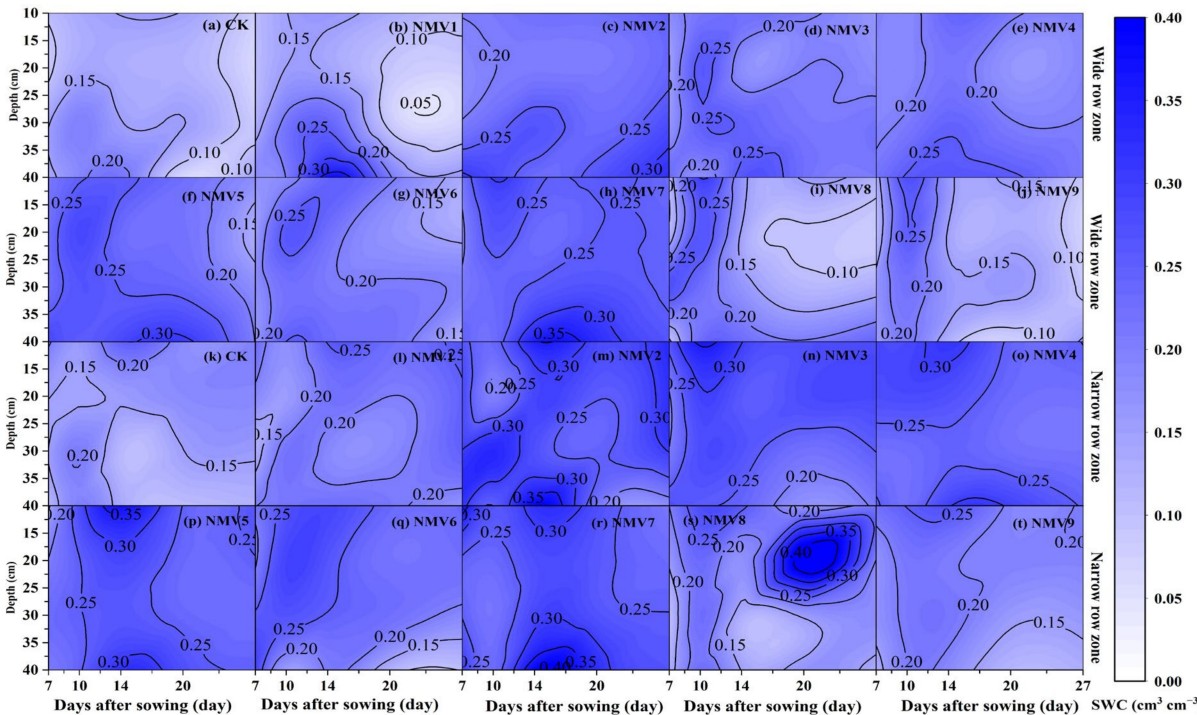

**Figure 7.** Distribution of soil water content in soil using film-mulched drip irrigation at dry sowing and wet emergence stages for the humic acid (HA), biochar-based microbial agent (M), and vermicompost (V) synergistic effect in 2022.

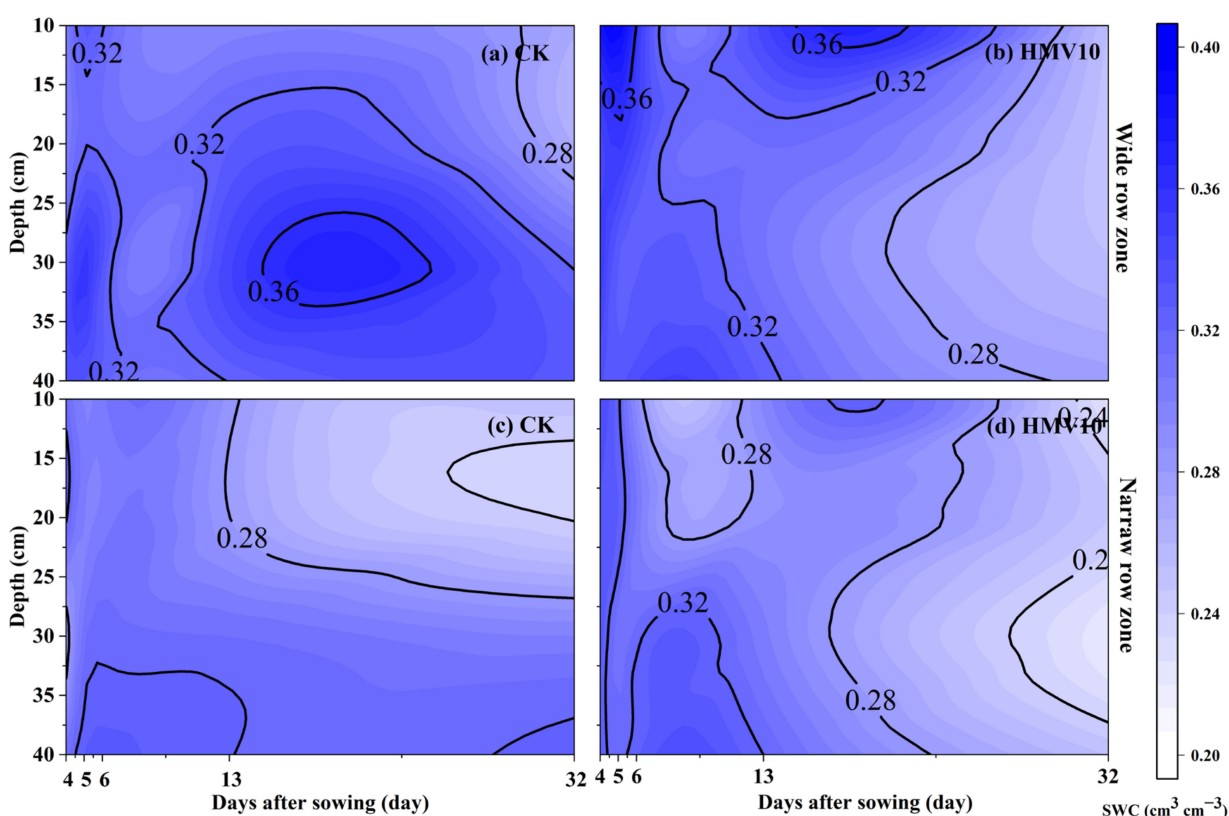

**Figure 8.** Distribution of soil water content in soil using film-mulched drip irrigation at dry sowing and wet emergence stages for the humic acid (HA), biochar-based microbial agent (M), and vermicompost (V) synergistic effect in 2023.

### 3.2.2. Soil Temperature

Under film-mulched drip irrigation conditions in 2021, 2022, and 2023, compared with CK, the application of HA, M, and V increased the ST of the 0–25 cm soil layer at the dry sowing and wet emergence stages of cotton by 6.1–17.8%, 1.7–7.7%, and 6.7% (Figure 9). The application of HA, M, and V found that the synergistic effect of the three soil amendments was better in 2021 (Figure 9a). In 2022, the ST in the 0–25 cm soil layer treated with HMV6 was the highest under the synergistic effect of the soil amendments (Figure 9b). In 2023, the ST increased significantly under the synergistic effect of the soil amendments (Figure 9c). The results showed that the applications of HA, M, and V soil amendments could all increase ST, and their coordinated application had a better effect. It increased the ST before cotton emergence, which was beneficial to cotton emergence.

### 3.2.3. Soil Salt Content

Under film-mulched drip irrigation conditions in 2021, 2022, and 2023, compared with CK, the 0–40 cm SSCs of the HA, M, and V treatments were reduced by 9.7–61.5%, 13.0–72.5%, and 8.9–46.6% (Figure 10). The SSC in the 0–40 cm soil layer was the lowest in the coordinated application of HA, M, and V in 2021. The SSC in the 0–40 cm soil layer was the lowest for the HMV4 treatment, and the SSC in the 30–40 cm soil layer was the lowest for the HMV1 treatment in 2022. The SSC of the 0–30 cm soil layer was significantly reduced in the coordinated application of HA, M, and V in 2023. The results showed that the soil amendments improved the SSC of cotton during the dry sowing and wet emergence stages, alleviated the inhibitory effect of salt on cotton emergence, and improved soil fertility. The collaborative application of HA, M, and V had better improvement effects.

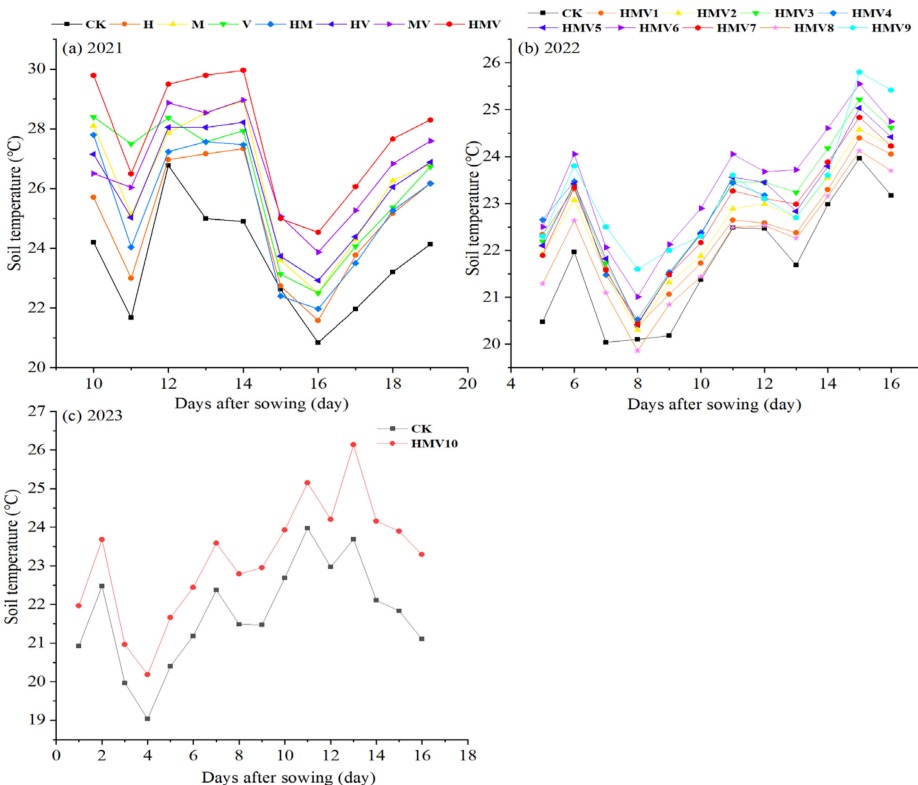

**Figure 9.** Effects of humic acid (HA), biochar-based microbial agent (M), and vermicompost (V) treatments on soil temperature at 0–25 cm at the dry sowing and wet emergence stages using film-mulched drip irrigation in 2021, 2022, and 2023.

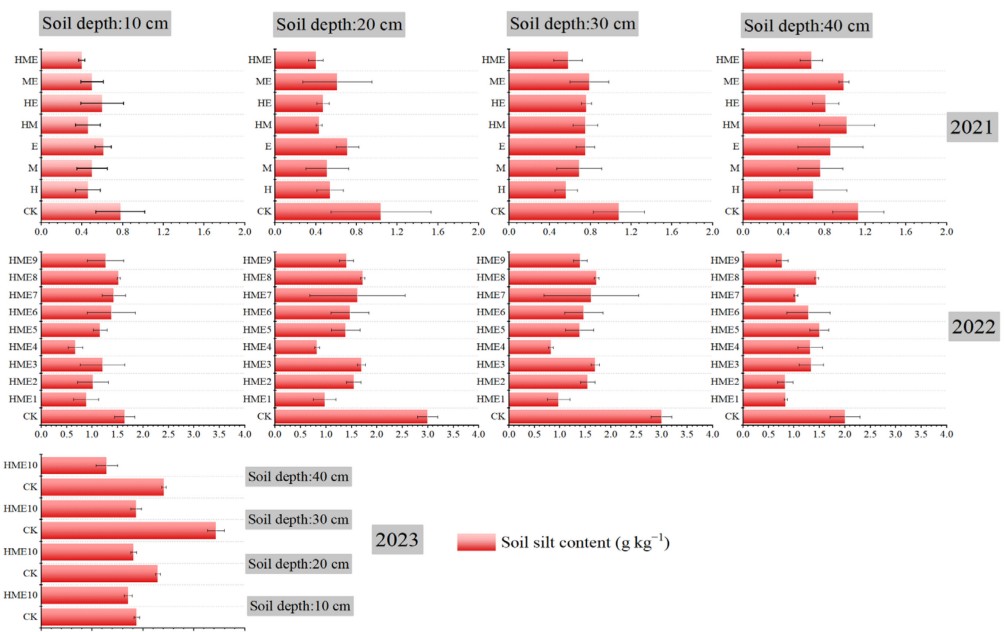

**Figure 10.** Effects of humic acid (HA), biochar-based microbial agent (M), and vermicompost (V) treatments on soil salt content at 0–40 cm for drip-irrigated cotton emergence using film-mulched drip irrigation in 2021, 2022, and 2023. Error bars represent standard errors.

However, comparing the SSC of the 0–40 cm soil layer in 2021, 2022, and 2023, it was found that the SSC increased. The main reason was that although the application of HA, M, and V in the soil reduced SSC, the long-term drought and lack of rainfall in Xinjiang caused SSC to migrate to the surface as water evaporates. At the same time, when the dry

sowing and wet emergence technology was implemented in the experimental area, winter irrigation and spring irrigation were not carried out to rinse the soil salt, which resulted in the accumulation of salt on the soil surface.

### 3.3. Effects of HMV Application on Soil Compaction Degree

There were significant differences in SCD in the 0–20 cm soil layer in 2021 and 2022, while no significant differences were observed between soil amendment treatments in 2021 (Figure 11). Compared with CK, the SCD of the 0–20 cm soil layer treated by HA, M, and V in 2021, 2022, and 2023 decreased by 30.6–46.0%, 10.6–44.7%, and 44.2%, respectively. The results showed that the soil amendments reduced SCD; the HMV treatment had the lowest SCD at 130 kPa in 2021, and the SCD treated by HMV9 was the lowest at 128.6 kPa in 2022. In 2023, the SCD improvement effect was obvious in the coordinated application of HA, M, and V.

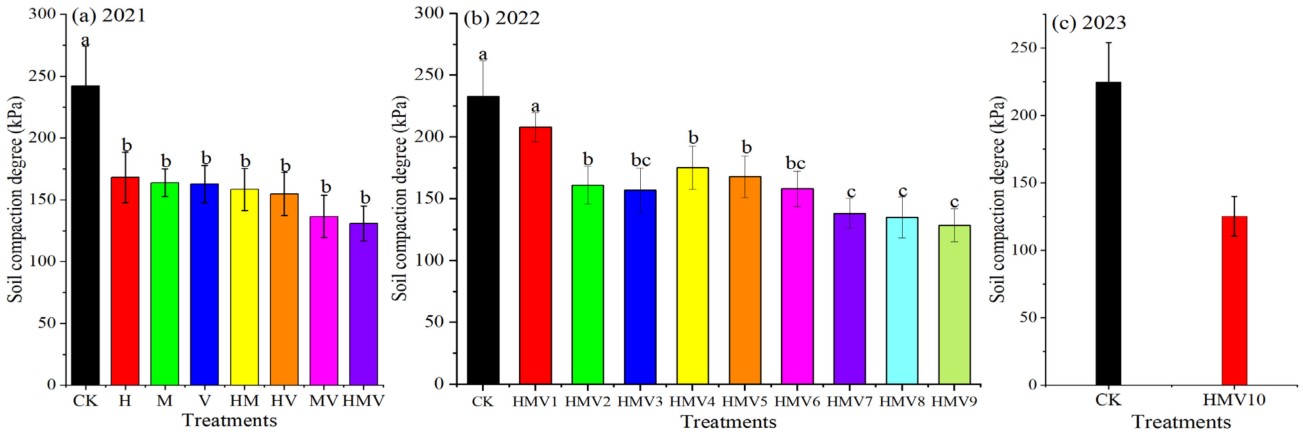

**Figure 11.** Effects of humic acid (HA), biochar-based microbial agent (M), and vermicompost (V) treatments on soil compaction degree at 0–20 cm in dry sowing and wet emergence stages of cotton using film-mulched drip irrigation in 2021, 2022, and 2023. Error bars represent standard errors. Different letters above the bars indicate statistical differences among treatments at the significance level of $p < 0.05$ using the LSD test.

### 3.4. Effects of HMV Application on Cotton Seedling Emergence Rate and Yield

There were significant differences in cotton emergence rate and yield among various soil improvement treatments in 2021 and 2022 (Figure 12). Compared with the control, the single and combined application of HA, M, and V in 2021, 2022, and 2023 significantly increased the cotton seedling emergence rate, increasing by 7.6% to 22.8%, 1.9% to 19.1% and 20.2%, respectively. The cotton yield in the single and combined applications of HA, M, and V in 2021, 2022, and 2023 were significantly increased, increasing by 7.0–31.8%, 8.1–54.1%, and 42.7%, respectively. In 2021, the seedling emergence rate and yield were the highest in the combined application of HA, M, and V, which were 22.8% and 5.80 t ha$^{-1}$, respectively. In 2022, the HMV9 treatment had the highest seedling emergence rate, which was 88.7%, and the HMV7 treatment had the highest yield, which was 7.51 t ha$^{-1}$. In 2023, the cotton seedling emergence rate and yield for the combined application of HA, M, and V were 85.7% and 7.08 t ha$^{-1}$, respectively. The cotton emergence rate and yield for three consecutive years were significantly higher than those in the control group, indicating that the separate application of HA, M, and V could also promote cotton emergence and increase yield, and the coordinated application of HA, M, and V had the best effect.

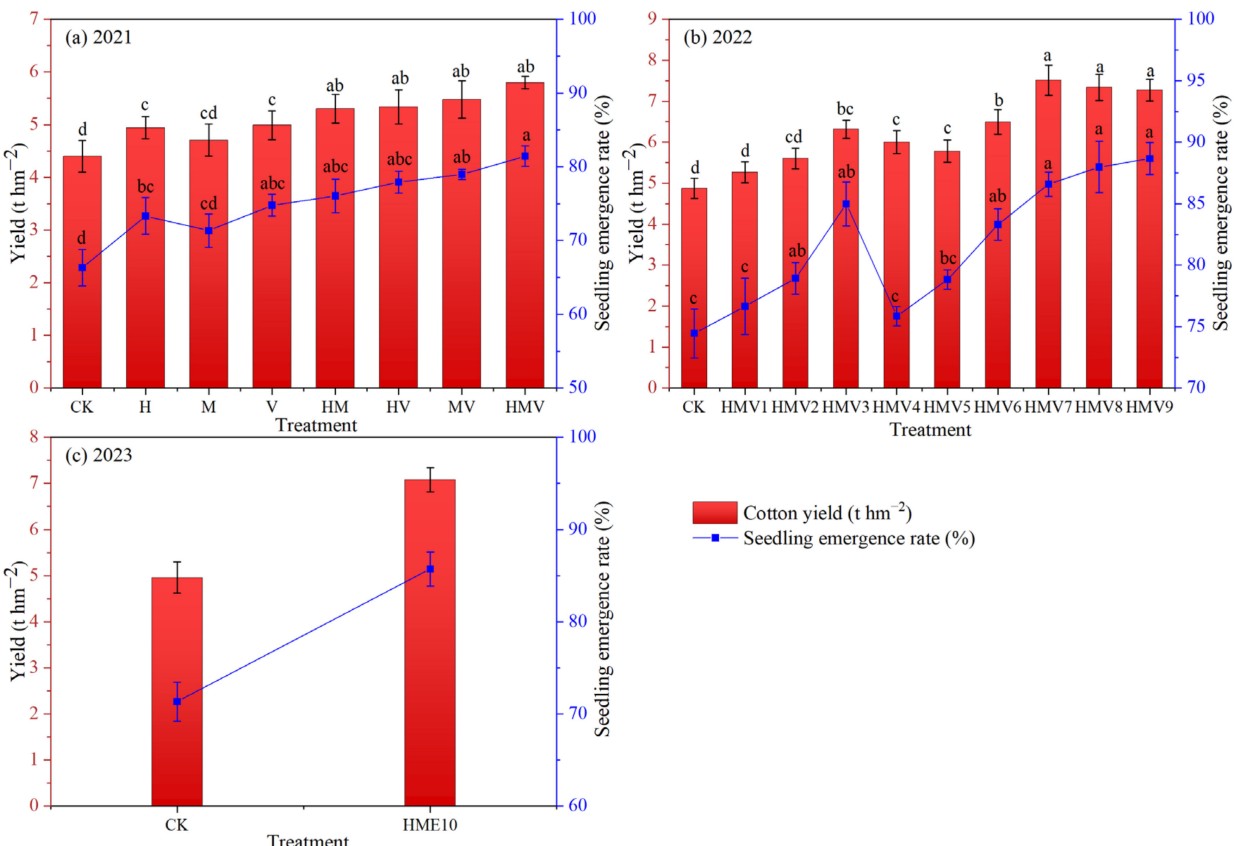

**Figure 12.** Effects of humic acid (HA), biochar-based microbial agent (M), and vermicompost (V) treatments on seedling emergence rate and yield of cotton using film-mulched drip irrigation in 2021, 2022, and 2023. Error bars represent standard errors. Different letters above the bars indicate statistical differences among treatments at the significance level $p < 0.05$ using the LSD test.

### 3.5. Correlation Analysis of Cotton Seedling Emergence Rate and Soil Indicators

A Pearson correlation analysis was performed between CY and SBD, SP, SWC, ST, SSC, SCD, and CY. The results showed that changes in cotton yield were mainly driven by comprehensive changes in soil property indicators. The order in 2021 was ST > SP > SWC > SSC > SBD > SCD (Figure 13a). Cotton yield had a very significant positive correlation with SER ($p < 0.01$), a significant positive correlation with SP ($p < 0.05$), and a positive correlation with ST and SWC. Cotton yield was significantly negatively correlated with SSC and extremely negatively correlated with SBD and SCD ($p < 0.01$). The order in 2022 was SER > SP > SWC > ST > SSC > SCD > SBD (Figure 13b). Cotton yield had a very significant positive correlation with SER, SP, and SWC ($p < 0.01$), a positive correlation with ST, a very significant negative correlation with SBD and SCD ($p < 0.01$), and a negative correlation with SSC.

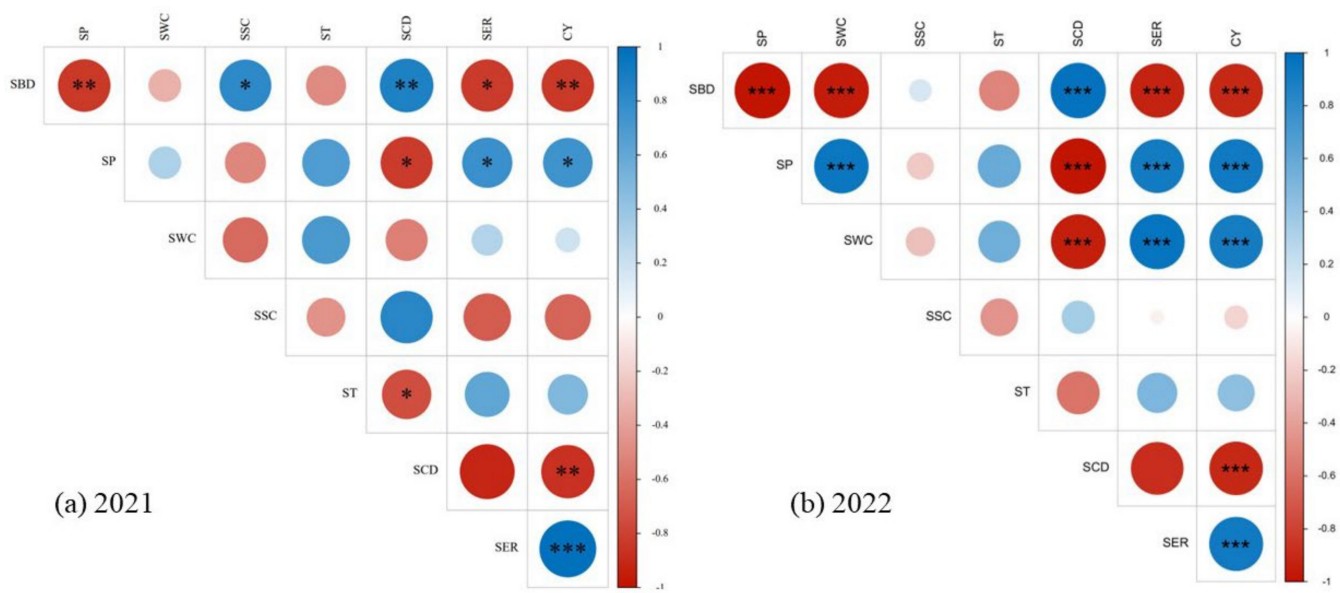

**Figure 13.** The influence of soil properties on cotton emergence rate was examined through correlation heat maps. Note: SBD, SP, SWC, SSC, ST, SCD, SER, and CY, respectively, represent soil bulk density, soil porosity, soil water content, soil salt content, soil compaction degree, cotton seedling emergence rate, and yield. *, **, and *** indicate significant effects at the 0.05, 0.01, and 0.001 probability levels.

## 4. Discussion

At present, many strategies have been proposed, such as winter irrigation, spring irrigation, and double-layer mulching, to ensure the emergence of cotton seedlings [32–35]. However, these strategies may cause water waste and increase mulch pollution [3], while HA, M, and V will not. Soil problems and cotton emergence were significantly improved after single/combined applications of HA, microbial inoculant, and V in this study. Soil physics, water–heat–salt distribution, and compaction are important indicators of soil fertility and health and can be used to evaluate crop growth [36,37]. Single and combined applications of HA, M, and V affected SBD, SP, SWC, ST, SSC, and SCD, which directly and indirectly promoted cotton emergence, and increased yield in Xinjiang.

### 4.1. Effects of Applying HA, M, and V on Soil Physical Properties

Soil compaction, salinization, and nutrient imbalance are the main factors affecting cotton seedling emergence [6,38]. In this study, the single and combined HA, M, and V applications reduced SBD and increased SP, among which the combined application of HA, M, and V produced the lowest SBD and the highest SP (Figure 4). This is due to the fact that after HA is added to the soil, loose soil particles are gathered together through flocculation, which regulates the bulk density of the soil, increases the total porosity of the soil, and forms a mass structure with good water stability [24]. Xu and Liu [39] showed that HA could increase the total number of soil aggregate structures by 1.5–3 times, and water-stable aggregate structures by 8.5–30%. The water, fertilizer, gas, and heat status in the soil layer, that is, the soil fertility, was significantly improved. Liu et al. [15] have shown that HA has colloidal properties and can form aggregates with soil aggregates, thereby reducing SBD and increasing porosity. Secondly, M has the characteristics of a porous biochar structure [19], which can reduce SBD and increase soil porosity [31]. V is porous, lightweight, and contains rich organic matter. Therefore, applying V to the soil can improve the soil structure, effectively reduce the SBD, and increase soil porosity [40,41]. The nutrient content of HA is higher than that of M and V, which can promote soil microbial activity and increase the content of beneficial microorganisms in the soil [42,43]. The loose and porous characteristics of V and M can provide a suitable living space for microorganisms, which is conducive to microbial reproduction, promotes their metabolic activities, and

activates fixed nutrients in the soil [19,41], and M can maintain microbial activity and protect microorganisms from adverse environmental factors [44]. Therefore, the combined applications of HA, M, and V had a greater improvement effect on soil bulk density and porosity than the single HA, M, and V applications.

In addition, this study found that the single or combination application of HA, M, and V reduced soil compaction, and soil compaction had a significant positive correlation with soil bulk density and a significant negative correlation with soil porosity (Figure 11). Yang et al. [45] reported that in soils with the same or similar textures, bulk density and porosity can well reflect the degree of soil compaction. Similar results were also reported by Al-Shatib et al. [46], who stated that soil compaction leads to an increase in soil dry bulk weight and seepage barrier capacity. Therefore, the effect of soil amendments on soil compaction is attributed to changes in SBD and porosity.

### 4.2. Effects of Applying HA, M, and V on Soil Water, Heat, and Salt

Many studies have shown that the application of HA, M, and V can effectively improve the hydrothermal and salt environments of the soil [47–50]. Wang et al. [51] found that the application of HA and a microbial agent to soil can increase the water content of saline soil and inhibit soil salinity. Zheng et al. [52] found that the application of HA fertilization in wet irrigation is beneficial to increasing the surface soil temperature. Li's [53] research shows that applying HA to farmland soil can promote salt leaching in the surface soil, thereby reducing soil salinity. In addition, the application of V to soil can significantly increase SWC, reduce SSC, and eliminate the adverse effects of soil salinization and water stress [54]. In this study, similar results were obtained with single and combined applications of HA, M, and V.

This might be attributed to the following reasons: (i) HA is mainly composed of carbon chains with an aromatic structure and a variety of highly active chemical functional groups, and its specific surface area is huge, up to 2000 $m^2\ g^{-1}$. Hydrophilic groups such as alcohol hydroxyl groups, phenol hydroxyl groups, carboxyl groups, and carbonyl groups on HA ionize and combine with water molecules to form hydrogen bonds after contact with water, absorb water, and increase soil moisture content [24]. (ii) HA is a macromolecular organic amphoteric substance, and its acidic functional groups can release $H^+$ to neutralize and react with alkaline substances in soil to produce $H_2O$, reducing the alkalinity of soil. Functional groups such as aldehyde groups and carboxyl groups in HA can form humate with cations in soil, forming a buffer system for the mutual conversion of HA to humate. At the same time, it can also combine with various cations in the soil, so that the harmful ions in the soil solution can exchange with HA and reduce the soil salt content [24]. (iii) An M can promote microbial activity and increase soil microbial activity, which can increase soil permeability and be beneficial to the soil hydrothermal environment. Moreover, the biochar in M has the ability to adsorb and fix salt, and can absorb salt in the soil, reducing the accumulation of salt on the soil surface [19]. (iv) The application of V to saline soil increases the contents of exchangeable potassium ions, calcium ions, and magnesium ions, and decreases the contents of exchangeable sodium ions in the soil, thus reducing the salinity of the soil [55]. Secondly, V is rich in organic matter, which can increase soil organic matter content, improve soil fertility and water retention capacity, and reduce water evaporation and salt accumulation in the soil [56]. In addition, the organic matter and microorganisms in V can promote the formation of soil aggregates and increase soil permeability [40], which is beneficial to soil warming under film-mulching conditions at the cotton seedling stage in Xinjiang.

In addition, some studies have shown the combined application of soil amendments has an impact on soil water, heat, and salt. Research by Han et al. [30] showed that the combined application of biochar and HA has a more obvious impact on soil electrical conductivity and soil nutrients. Liu et al. [15] and Wang et al. [57] found that the combined application of HA and V can increase soil aggregate content and effectively reduce soil salinity. The results of this study show that combined soil amendments can improve the

problem of soil salt accumulation on the surface caused by dry sowing and wet emergence technology, but the long-term implementation of dry sowing and wet emergence technology will still cause soil salinization.

### 4.3. Effects of Applying HA, M, and V on Cotton Seedling Emergence and Yield

The dry climate conditions and long-term mulching of cotton fields in Xinjiang will reduce fungal diversity in the soil and accumulate soil salt on the surface, posing potential risks to the ecological environment of farmland soil [58,59]. Previous studies have proven that the application of HA, M, and V can improve the soil microbial environment, reduce soil salinity, and promote the growth of cotton seedlings [60–62]. However, it is unclear whether the mixed application of HA, M, and V can produce synergistic benefits for cotton seedlings, especially in mitigating the surface accumulation of soil salt in Xinjiang.

The combined application of soil amendments increases soil aggregate content, promotes soil microbial activity, and improves crop growth and nutrient absorption [15,63]. According to the correlation analysis results, SP, SWC, and ST have positive effects on cotton emergence, while SBD, SSC, and SCD have negative effects (Figure 11). HA, M, and V contain a large amount of nutrients and micronutrients, increasing soil organic matter content [19,26,63,64]. In addition, HA and V are rich in different types of growth hormones, which promote crop growth [65]. This also explains why in this study, the HA and V treatments performed better than M in improving cotton emergence rate and yield. However, the synergistic effect between HA, M, and V is currently unclear. In this study, it was observed that the single applications of HA, M, and V were beneficial to cotton seedling emergence and increased production, and the synergistic effect of HA, M, and V was better (Figure 10).

### 5. Conclusions

It was found that the single and combined application of HA, M, and V improved soil physical properties and water–heat–salt distribution and increased the emergence rate and yield of cotton. Soil bulk density, soil salinity, and soil compactness were reduced by 2.2–20.8%, 8.9–46.6%, and 10.6–46.0% on average when treated with HA, M, and V. Soil porosity, soil water content, soil temperature, and cotton emergence rate increased by 4.4–87.1%, 2.0–45.9%, 1.7–17.8%, and 1.9–22.8%. The combined application of HA, M, and V ($60 \, \text{kg ha}^{-1}$, $120 \, \text{kg ha}^{-1}$, and $120 \, \text{kg ha}^{-1}$) is recommended to promote the implementation of dry seeding and wet extraction techniques in salinized cotton fields in Xinjiang to alleviate the pressure caused by water shortage in Xinjiang. However, the amount of soil amendments applied in this study was progressively refined over three years to determine the optimal combination of application, which led to a lack of replication of the trial results, making the conclusions limited. Secondly, the study did not consider the effects of HA, M, and V synergy on soil evaporation, nutrient transport, or fertilizer finiteness. Therefore, these studies need to be carried out in the future.

**Supplementary Materials:** The following supporting information can be downloaded at: https://www.mdpi.com/article/10.3390/agronomy14050994/s1, Figure S1: The relationship between the values of $EC_{1:5}$ and soil salt content; Figure S2: Effects of humic acid (H), biochar-based microbial agent (M), and vermicompost (V) treatment on soil water content at 0~40 cm in dry sowing and wet emergence stages under film drip irrigation in 2021; Figure S3: The synergistic effect of humic acid (H), biochar-based microbial agent (M), and vermicompost (V) on the 0–40 cm soil water content in the dry sowing and wet emergence stages of drip irrigation under the film in 2022; Figure S4: The synergistic effect of humic acid (H), biochar-based microbial agent (M), and vermicompost (V) on the 0–40 cm soil water content in the dry sowing and wet emergence stages of drip irrigation under the film in 2023.

**Author Contributions:** Conceptualization, Y.S., Y.B., and H.L. (Hongbo Liu); formal analysis, G.L.; funding acquisition, J.Z. and Y.S.; investigation, W.N., J.Z., H.L. (Hongbo Liu), Y.D., X.W., and H.L. (Hongqin Lu); methodology, Y.B., Q.W., J.Z., H.L. (Hongbo Liu), and H.L. (Hongqin Lu); supervision,

Y.B.; validation, Q.W. and X.W.; visualization, G.L.; writing—original draft, Y.D.; writing—review and editing, G.L. All authors have read and agreed to the published version of the manuscript.

**Funding:** This research was supported by the Major Special Science and Technology Project of Xinjiang Province (2023A02012-1), the Xinjiang Tianshan Talent Leader-ship Training Project (2022TSY-CLJ0069), the Key Research and Development Program of Shaanxi (Program No. 2024NC-YBXM-243).

**Data Availability Statement:** Data will be made available on request.

**Conflicts of Interest:** The authors declare that they have no competing interests.

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
