# Peer review of "Synergistic Effects of Humic Acid, Biochar-Based Microbial Agent, and Vermicompost on the Dry Sowing and Wet Emergence Technology of Cotton in Saline–Alkali Soils, Xinjiang, China"

_agronomy, doi:10.3390/agronomy14050994_

Round 1

Reviewer 1 Report

Comments and Suggestions for Authors

Dear Editor and reviewers, the article has the potential to be published and needs several improvements, as highlighted below:

- In the introduction to the abstract, authors must highlight in one sentence that vermicomposts also contain humic acids.

- In the introduction, some paragraphs are huge, see the first paragraph, please reduce it or divide it into two.

- In line 60, authors must insert a sentence showing that despite the cultivation limitations highlighted in the first paragraph, the use of humic acid, Biochar-based microbial agent and vermicompost can improve the conditions of plants to withstand these stresses, optimizing productivity. There are several works in this line of research.

- I suggest changing the abbreviation of humic acids from H to HA throughout the article, it is very common in all literature to use the acronym HA.

- In line 61-62: Humic acid (H) is a macromolecular substance that constitutes soil humic substance and has been widely used in many fields. Currently the most accepted theory is that HÁ are a supramolecular structure, in addition, HÁ can be extracted from various materials such as the vermicompost used, so I suggest changing and inserting this information.

- In line 66: As a condition soiler, humic acid can also change soil texture, nor cultivation conditions generally change the texture of the soil, as the simple application of HÁ could promote such an effect, the information is incorrect, I suggest removing it. Furthermore, few mechanisms for the use of HA have been explored, the authors do not even talk about its bioactivity, I suggest reading and condensing the effects mentioned in this review article (https://doi.org/10.3390/molecules26082256).

- Line 79 is not PH value but pH value.

- Line 87-95, the authors mention the effects of vermicompost, but a part of its effects is related to its content of humic substances such as humic acids, the authors should mention this, and how the effect of vermicompost can be related to the part related to nutrient content, or the HAs contained therein.

- Line 96-107: this interaction is poorly explored, the authors need to succinctly give more details of the chemical principles involved in this interaction, highlighting that some materials are rich in some compounds and promote this effect, while other materials are rich in other compounds and have other effects.

- The quality of Figure 2 is poor, especially the MAP, please improve it.

- Section 2.2.1. Sources of Humic Acid, Biochar-Based Microbial Agent and Vermicompost is very poor, more details must be provided on the place of origin of each product, the process of obtaining it, as this will affect the properties of the materials and their effects on the soil and on plants, in addition, there is a need for a physical-chemical characterization of the materials tested, including nutrient content, pH, electrical conductivity, etc. Because this way it is possible to establish a cause and effect relationship, without characterizing the materials used, the experiment simply seems like a product test. Provide a paragraph for each material, highlighting its origin and obtaining process, place of production, and physical and chemical properties.

- Because the dose of fertilizers varied according to the year, the authors need to establish a scientific basis for this, as the way it is written it seems like it was an idea that came out of nowhere. Treatments do not follow an application logic.

- The material and methods section needs to be better described, highlighting the collection process and methods used.

- The quality of figures 5 and 6 needs to be improved

- In figure 12, invert the colors, placing red for negative correlations and blue for positive ones.

- Please remove subtopics from the discussion and make it more fluid.

- In line 384 the authors mention that the nutrient content may be high in HA, but how can they say this if they have not adequately characterized the materials tested, because HA and its amount of nutrients is very variable. Once again, I emphasize that authors must provide information on the characterization of the materials used

- In the discussion, authors need to show the effects and explain the mechanisms involved, they can improve and deepen the discussion of the article.

- The conclusion must be redone according to the objectives of the article, as it is written it looks like a summary of the results section

Reviewer 2 Report

Comments and Suggestions for Authors

Synergistic effects of humic acid, biochar-based microbial agent and vermicompost on the dry sowing and wet emergence technology of cotton in Xinjiang, China

This study focuses on the application of Soil amendments such as humic acid (H), biochar-based microbial agent (M) and vermicompost (V), which can improve soil quality and promote crop growth

1- In the title, I think it is better to add “in the saline-alkali soils” to clarify this study

2- In abstract, the authors used “the application amounts of H, M and V were 75 kg hm−2, 75 kg hm−2 and 225 kg hm−2” please use SI units not hm-2

3- In general, in keywords, please avoid to repeat any word already mentioned in the title

4- line 33 and 39 and many others in the MS, please follow the instruction of the journal

“China (Appiah et al., 2014). As the cotton plant-“, “fall (Pan et al., 2011).”, etc.

5- Introduction must include very update refs. mainly from 2024, 2023, 2022, and 2021, please

6- it is recommended please to add a flowchart including the main treatments, just mention the measurements, with supplying photos if that possible especially, that we have a different experimental Design in each year, along with Table 2 and Figure 3 as well

7- Why the authors used soil depth till 100 cm, are these soil amendments can be effective within this long depth, please clarify?

8- Please use the common soil term soil salinity (EC in ppm or dS m-1), not Soil salt (g·kg−1 ) in Table 1

9- How did the authors apply these treatments? Please clarify are these within drip irrigation? Or which method please, which volume per each plot for each treatment

This is not clear:

“Spread H, M and V evenly on the soil surface of each plot, mix them thoroughly with the soil with a wooden rake, and then use a rotary tiller to plough the soil to 30 cm from the surface. The emergence water was poured in once, and the water volume for emergence was 900 m3 hm−2”

10- Where the characterization of soil used “the saline-alkali soils” in section of Materials, please

11- Please this section “2.3.5. Statistical analysis” must be in plural “analyses”

12- Figures 5, 6, and 7 need to be clear in a high resolution, please

Comments on the Quality of English Language

moderate 

Round 2

Reviewer 1 Report

Comments and Suggestions for Authors

Dear Editor and authors, the manuscript was improved and can be accepted in the current version.